# Human knowledge models: Learning applied knowledge from the data

Egor Dudyrev[1], Ilia Semenkov[1,2], Sergei O. Kuznetsov[1], Gleb Gusev[2], Andrew Sharp[3], Oleg S. Pianykh[3,4]*

1 HSE University, Moscow, Russia, 2 Artificial Intelligence Research Institute, Moscow, Russia, 3 Medical Analytics Group, Massachusetts General Hospital, Boston, Massachusetts, United States of America, 4 Harvard Medical School, Boston, Massachusetts, United States of America

* opianykh@mgh.harvard.edu

## Abstract

Artificial intelligence and machine learning have demonstrated remarkable results in science and applied work. However, present AI models, developed to be run on computers but used in human-driven applications, create a visible disconnect between AI forms of processing and human ways of discovering and using knowledge. In this work, we introduce a new concept of "Human Knowledge Models" (HKMs), designed to reproduce human computational abilities. Departing from a vast body of cognitive research, we formalized the definition of HKMs into a new form of machine learning. Then, by training the models with human processing capabilities, we learned human-like knowledge, that humans can not only understand, but also compute, modify, and apply. We used several datasets from different applied fields to demonstrate the advantages of HKMs, including their high predictive power and resistance to noise and overfitting. Our results proved that HKMs can efficiently mine knowledge directly from the data and can compete with complex AI models in explaining the main data patterns. As a result, our study reveals the great potential of HKMs, particularly in the decision-making applications where "black box" models cannot be accepted. Moreover, this improves our understanding of how well human decision-making, modeled by HKMs, can approach the ideal solutions in real-life problems.

**Data Availability Statement:** The data for E-, H-, and F- sets used in this study are available from standard public repositories (https://www.kaggle.com/adityakadiwal/water-potability, https://www.kaggle.com/datasets/S%C3%ADdrio-Libanes/

## Introduction

Artificial intelligence (AI), fortified by modern machine learning (ML), has demonstrated remarkable results in many areas of science and applied work, from ecology to healthcare. This progress has largely been driven by the development of complex ML models, capable of learning non-trivial patterns from large, multidimensional data.

This approach proved to be very fruitful in many areas of human knowledge except one: *human* knowledge itself. Known as "black boxes", most modern ML algorithms merely transform large volumes of original data points into equally large volumes of model coefficients and weights, without distilling any concise, conceptual knowledge that humans would be able to learn and to use. Moreover, standard machine learning results usually come in the same

covid19, and https://archive.ics.uci.edu/ml/
datasets/Polish+companies+bankruptcy+data#
respectively). The dataset for S-set, that we have
created ourselves, is provided with this paper as
Supporting information."

**Funding:** The author(s) received no specific
funding for this work.

**Competing interests:** The authors have declared
that no competing interests exist.

format of "optimal instance" solutions, tied to a particular instance of a trained model–and
therefore cannot be explained, generalized, scaled, or modified without rerunning the model.
This distinctly disconnects from the classical way of human learning, where the knowledge–in
the form of simple rules, laws, and actionable logic–is learned directly from the data by repeti-
tive trial-and-error. Consequently, this disconnect poses a number of fundamental barriers to
the use of the present ML models:

1. Black box AI cannot be used in areas where the final decisions ought to be made by
   humans–ranging from healthcare and education to management and complex industrial
   systems.

2. ML "optimal instance" solutions become highly impractical in applications with frequent
   data changes and drifts.

3. Complex ML models, requiring significant resources to develop, set up and maintain, are
   immediately ruled out in a large number of real-life applications where the necessary
   resources are not available or do not provide sufficient return on investment.

The challenge of converting data into simple and humanly comprehensible logic has been
addressed in many areas of previous research. In the early 1960s, work on formal logic led to
the inception of logical programming and rule-learning algorithms [1–8]. The latter–including
algorithms such as Corels [9], Slipper [10], Skope-Rules [11], RuleFit [12], LRI [13], MLRules
[14], and more–often rely on greedy approaches to extract short Boolean expressions from
more complex models (such as large decision trees). For example, rule induction can be done
by considering each single decision rule as a base classifier in an ensemble, which is built by
greedily minimizing a loss function (Slipper, LRI, MLRules); or by extracting the rules from an
ensemble of trees, and then building a weighted combination of these rules by solving an
L1-regularized optimization problem (RuleFit).

With the recent growth of real-life AI implementations, the concepts of "transparent",
"decomposable", "interpretable", or "explainable" AI have also become the focus of applied AI
research and analysis [11, 15–20] by either reducing more complex models to their simpler
versions, or by providing additional insights into the complex model functionality (such as fea-
ture importance and similar model explainers) [21–23]. In particular, *interpretable models* are
most commonly defined as the models where a human can understand the causes of model
prediction (e.g., linear regression), and *simulatable models* are those where a human can repro-
duce the model reasoning and results (e.g., decision-rule models) [11, 24].

Although helpful, these approaches run into a few principal limitations. First, instead of
explaining the original data, many of them focus on explaining the "black box" models, sec-
ondary to the original data [25, 26]. This leads to a number of "reality disconnects" including
incorrect data interpretations, inheriting model bias and false "shortcuts", lacking satisfactory
explanations, and more, resulting in significant criticism [27, 28]. Second, the key concepts of
"interpretability" are not clearly defined, leaving ample room for imagination [26, 29]. More-
over, they are treated very passively, as our ability to understand, but not to apply–thus ruling
out active, applied use of the model logic.

Many of these shortcomings originate from the "machine" approach to data exploration,
focused on building an optimal computer algorithm rather than extracting human-like knowl-
edge. While humans use their knowledge to conduct more experiments and to generate more
data, converting data back to more knowledge seems to be largely missing. There is a definite
need for building human-knowledge-extracting models, which would not only help us under-
stand the data in the most interpretable and actionable way but would also help us explore the
limits of human "understandability" for any given problem or dataset.

Therefore, in this paper, we would like to address these limitations by formalizing the class of mathematical models that an average human can easily evaluate mentally, without using any additional resources. We did this by starting from a vast volume of research in human cognitive sciences, thus introducing the class of "human knowledge models" (HKMs). Using this formalization, we discuss how one can learn HKMs directly from data, without relying upon any intermediate "black boxes." This enables us to compare the quality of HKMs to that of classical black boxes, using several datasets from different application areas.

## Methods and materials

### Human knowledge model class

By "human knowledge models" (HKMs) we understand the subset of functions which, once known, can be *evaluated by humans in real time*, *mentally*, *without using any additional resources*. Thus, we are primarily concerned with active, applied knowledge, as opposed to more passive "interpretable" or "explainable"; and we emphasize "fast and easy to compute" as opposed to "simulatable" or "humanly-reproducible" AI. Human ability to memorize and to compute has been a subject of extensive research in psychology and cognitive sciences [30–33], leading us to the following principal HKM constraints:

1. Boolean operators: OR, AND, NOT, and thresholded Boolean SUM (arithmetic sum of Boolean variables, compared to an integer threshold) [29, 34].

2. At most four (Boolean) variables, where each variable is used at most once [34–36].

3. At most four (Boolean) operators [35–37].

4. Simple numerical thresholds for converting non-Boolean variables into Boolean.

Note that the class of HKM models can intersect with many previously defined ML model classes but does not correspond to any of them exactly (Fig 1). For example, HKMs must be

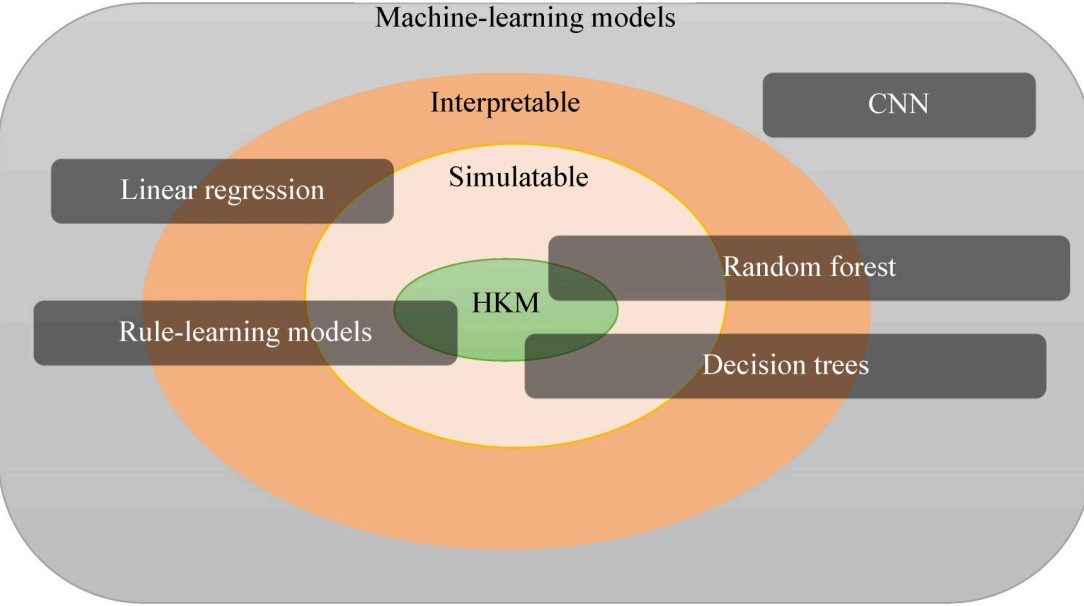

**Fig 1. HKM and machine learning.** HKMs can be viewed as more constrained forms of several classical ML models (such as logic rule-learning), but they will be distinctly different from any hard-to-compute ML (regression, CNN).

interpretable (as well as explainable or simulatable), but the opposite is not true: many classical interpretable models, such as linear regression, which requires many complex decimal multiplications, cannot be efficiently computed by humans. Similarly, the use of SUM for evaluating expressions such as "if two of the three conditions are met" (routinely used in human decision making) exceeds the branching logic of a decision tree but can be viewed as a small random forest of stumps.

## HKM model training

Training an HKM includes the following components:

1. Learning the best data thresholds (to convert the original features into Booleans)

2. Learning the best subset of features (up to 4 features used in the model)

3. Learning the best model logic (the choice of Boolean operators)

4. Selecting M best HKMs, achieving the highest fit

The implementation of this approach is shown in Fig 2. We begin by generating all possible Boolean functions of N variables (N≤4) in the form of truth tables. Then each table is converted to a DNF (disjunctive normal form), commonly used by many decision rule and decision set models. Therefore, we add a new step–*semantically-equivalent formula simplification*–to reduce DNFs to the shortest possible Boolean expressions (including thresholded SUM

<u>Input</u>: Dataset with $N$ variables, $L$ observations
<u>Model parameters</u>: Model size (variable count) $n \leq 4$, threshold count $K \approx 10$, number of best models to output $M$
<u>Output</u>: $M$ best HKM models (Boolean expressions) of $n$ variables

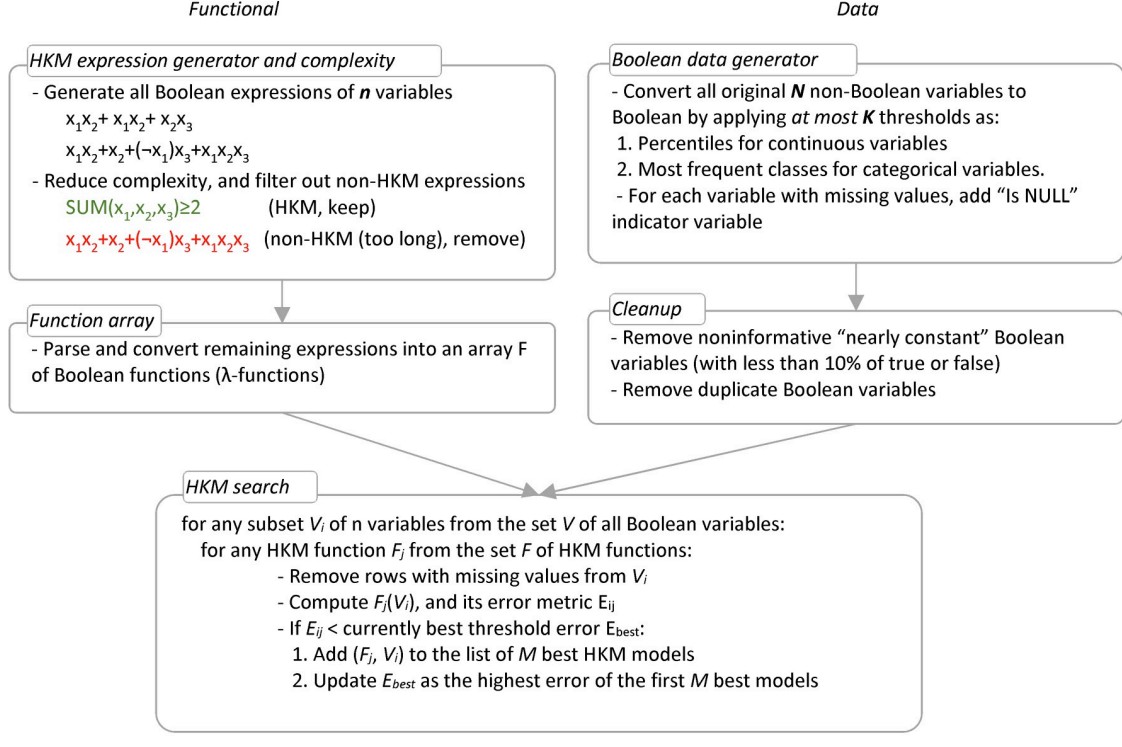

**Fig 2. HKM algorithm outline.** Note that the HKM search loop can be easily computed in parallel.

formulas), which preserving their meaning (truth table). This simplification is essential to ensure that the Boolean expression can be made simple enough to be efficiently remembered and processed by humans. If the complexity of the simplified expression still exceeds our HKM constraints stated above, the formula is excluded from further processing. The relatively small number of remaining formulas (functions) are applied to all subsets of N variables, drawn from the original variable set. By maintaining a stack of the M currently best (formula, variable subset) pairs $(F_j, V_i)$, the algorithm discovers the best HKMs at explaining the target outcome.

Thus, HKM training learns both the best features and the best Boolean expressions, by traversing all their combinations. This makes it a form of Boolean expression learning rather than more conventional "coefficient fitting" ML (Fig 2). However, this is exactly what we want to achieve: we want to find the best possible expressions of low computational complexity, which then can be used by humans to compute the optimal results on their own.

As a result of this HKM training approach, the principal difference between HKMs and the previous rule-learners lies in the fact that HKMs narrow down the class of acceptable rules but broaden the optimal feature selection by using exhaustive subset selection. This sets HKMs aside from both classical rule learners such as Skope-Rules or RuleFit, where the rules are greedily learned by using decision trees, and standard machine-learning feature selection, where the best features are identified by random search (random forest), or greedy feature selection (Lasso regression, stepwise selection, etc.). The tradeoff between the smaller HKM rule subset and larger feature search space works extremely well in the case of small variable counts, as we discuss below.

A few more practically important aspects of HKM learning include:

- *Preserving the missing values.* Missing values can be important and informative; therefore, we do not impute any of them. Instead, we encode them as indicator variables and omit the missing values only when they show in specific subsets $S_i$–where the probability of having incomplete rows is minimal.

- *Learning M best HKMs.* In classical ML, it is much more common to produce a single (best) model instance through fitting the coefficients (weights, thresholds) of a predefined ML function. In HKM learning however, the Boolean function itself becomes a subject of model training, and several different functions can achieve nearly optimal quality of their HKMs. Unusual at first, this however reflects the fact that in reality one can often arrive at the same optimal goal by different means. Therefore, we learn several top models (M≈10–50), and then let the user examine them and decide which one would be the most practical to use.

## Experimental data

To experiment with HKM learning, we have selected several datasets from different applied areas dependent on active human decision-making and responsibilities:

- *Healthcare (H-set)*: COVID-19 diagnosis dataset, to predict the need of intensive care units for patients with COVID-19. The dataset includes 1925 observations with 229 features [38].

- *Ecology (E-set)*: Water potability, to predict whether the water has drinking quality. The dataset includes 3276 observations with 9 features [39].

- *Finance (F-set)*: A dataset of corporate records, to predict company bankruptcy status after the first year (5910 observations with 64 features) [40].

- *Strategy/Gaming (S-set)*: NIM dataset, to predict the winner of the NIM game [41]. This set was generated by us, and it includes 2000 observations with 10 features.

The datasets were chosen to have at least 1000 observations and several features, to allow experiments with models of different sizes and complexities. All datasets except S-set had unbalanced Boolean outcomes (27%, 39%, and 7% respectively), and contained missing values–which were treated as such, without imputing, to preserve the realistic data.

To compare the class of HKM functions with classical machine learning algorithms, we used the selected datasets to train a variety of standard "black box" and rule learners, which we divided into three principal classes:

- *Simulatable AI*: Decision tree, Skope-Rules and RuleFit models, constrained to use at most 4 features, but as many rules as needed to achieve best model quality. These models incorporated Boolean logic, which can be easily evaluated by a human.

- *Interpretable AI*: Decision tree, Skope-Rules, RuleFit with any number of variables, as well as reduced versions of random forest, XGBoost and CatBoost of up to 4 tree stumps and up to 4 variables, and logistic regression. These models corresponded to standard machine learning models with either simple Boolean logic, or small numbers of variables, which can be still "understood" by a human.

- *Black box*: Full versions of random forest, XGBoost and CatBoost, where the best size of the model was determined only by its hyperparameter training and exceeded the size of easily understandable/interpretable.

For each model, we ran $N_{exp}$ individual model training experiments, $N_{exp}$ varying between 1000 and 5000 depending on the model training time. Since optimizing model hyperparameters for each model run was not feasible due to significant runtime, we optimized them by selecting 10 random subsets with 80% of the original data, using grid search with 5-fold cross-validation on each subset. For each model type, this produced 10 different sets of optimal hyperparameters, which then were used in $N_{exp}$ experiments, randomly selecting an optimal hyperparameter set for each model training. For models depending on random seeds, the seeds were randomized at each experiment as well. Once optimal hyperparameters were determined, all models, including HKMs, were trained using standard 80/20 train/test splits and 5-fold cross-validation.

Using this multi-instance training approach, for each type of classical "black box" model, we obtained $N_{exp}$ best model instances, to represent the highest model performance within each model class. These best instances were compared with the best HKMs, using different metrics of model performance ($F_1$ was chosen as a default), and four different sizes of HKM models (HKM variable count *n*, increasing from 1 to 4).

## Results and discussion

### Model quality

Undoubtedly, HKMs achieve human computability through a significant reduction in model complexity, so one would expect a corresponding reduction in the model quality as well. Therefore, we studied the distribution of $F_1$ values, obtained from $N_{exp}$ = 1000 model instances trained on our four datasets. We used HKMs of size 4 (using 4 features) and compared them to a decision tree model (as the classical rule-learning model, also used in many rule-learning algorithms), and XGBoost (as a more complex ensemble model, which also achieved the highest quality on our data). We did not constrain the sizes of decision tree and XGBoost, so both models acted as "black boxes". The results of this comparison are presented in Fig 3 as histograms of $F_1$ values, and lead to several important observations.

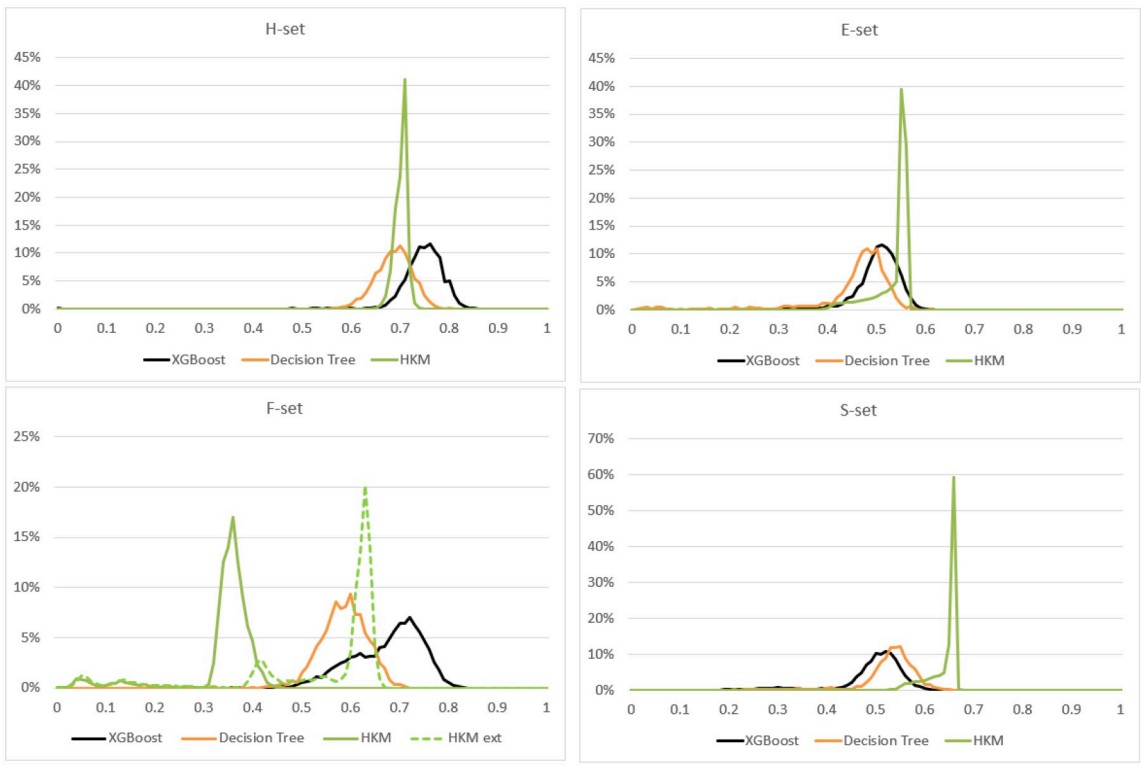

**Fig 3. Distribution of $F_1$ values for HKM, decision tree, and XGBoost models on four different datasets where they were trained.**

As one can observe by comparing the means of the histograms, HKMs outperformed "black boxes" on two of the four datasets (S- and E- sets). On H-Set, top HKM models scored slightly better than large decision trees on average, but approximately 5% lower than XGBoost (comparing the peaks of the two histograms). In practical applications, this 5% would present a very negligible difference, usually consumed by data and implementation noise. We can therefore conclude that HKMs performed roughly as well as "black boxes" on H-Set.

Consequently, as the F1 values in Fig 3 demonstrate, it was only the F-set where HKMs fell visibly behind the "black box" models. Therefore, we investigated further, and found that HKM performance was sharply improved when we added one more engineered feature–the product of two specific features (X32 and X33) from the original dataset. Then the resulting "HKM ext" outperforms decision trees, and approaches XGBoost quality (F-set, Fig 3). Thus, in the case of F-set, the predictive target was highly dependent on the feature multiplication– which HKMs could not reproduce by definition, since humans cannot compute real number products efficiently.

Thus, our numerical experiment in Fig 3 shows that HKMs can attain a rather outstanding model quality, to the point of outperforming the "black boxes". The latter sounds particularly surprising, but can be attributed to the following factors:

1. *Large HKM search space*. While each HKM is very small, enumerating all of them exhaustively creates a huge pool of models to choose from. For example, considering all 3-variable models from a set of 300 features already results in 4,455,100 possible variable subsets. This large number is further multiplied by the count of admissible 3-variable Boolean expressions (usually a few dozen, due to HKM constraints), significantly increasing the chances of finding a very predictive model.

2. *Intrinsic dataset dimensionality*. Despite large variable counts, many real-life datasets have much smaller "natural dimensionality", meaning that only a few variables are needed to explain the main data variability (a phenomenon frequently observed with principal component analysis). In this case, increasing model complexity beyond the natural dimensionality limits does not produce any significant gains, but increases the probability of not finding the best models. Moreover, models with significantly more degrees of freedom are more likely to deviate from finding general dependencies and overfit.

3. *Local minima and greedy optimization*. Most of the current "black box" machine learning relies on greedy optimization algorithms, applied to complex multidimensional functions with large number of local minima. This form of optimization, unlike exhaustive HKMs, does not guarantee finding the best (globally-optimal) solution.

It is also important to note that, while "black box" results were obtained with their greedy algorithms and probabilistic feature selection approaches, HKM exhaustive training is deterministic, producing the same best models from the same input dataset. That means that, at least on the same training data, we will always know the best HKM models, and therefore can compare only these top models to the less certain outcomes of "black box" training. This means that in Fig 3, we should be looking at the rightmost tails of the HKM histograms (best models), rather than the histograms' centers–which leads to even more favorable HKM performance.

## Model stability and multiplicity

In real life applications, achieving a good model quality "on average" is not enough: it is equally critical to ensure that the model performs consistently, with minimal deviation in its quality. Our experiment in Fig 3 reflects this important property in the spread of histograms, wider histograms corresponding to less consistent models.

To study this further, we quantified the *model stability factor* as the absolute relative change in model quality metric ($F_1$ in our case), taken between train and test sets:

$$S = \frac{|\text{Quality}(train) - \text{Quality}(test)|}{\text{Quality}(train)} = \frac{|F_1(train) - F_1(test)|}{F_1(train)}$$

Lower S values correspond to more stable models, and one can compute the probability of a model to remain stable below a certain threshold S as:

$$P(S) = Prob\left(\frac{|F_1(train) - F_1(test)|}{F_1(train)} < S\right)$$

Ideally, one would like $P(S)$ to approach 100% for the smallest values of S, but we computed the exact model stability curves for all major classes of ML models we investigated, using the four datasets (Fig 4).

Fig 4 confirms the expected trend: model stability generally increases in less complex models. However, HKMs clearly stand aside as the most stable, and significantly improving over "black boxes." First, this makes HKMs a more attractive choice for any practical AI implementations, when reliable model quality is a must. Second, it improves the model training process as well: HKMs are virtually impossible to overfit, remaining resistant to noise and bias challenges, so widespread with conventional AI. Consequently, this makes HKMs more appealing for continuous learning AI, when one must ensure that model retraining will still produce a reliable result. Finally, this also leads to better model scalability and generalizability, so appreciated in practical applications.

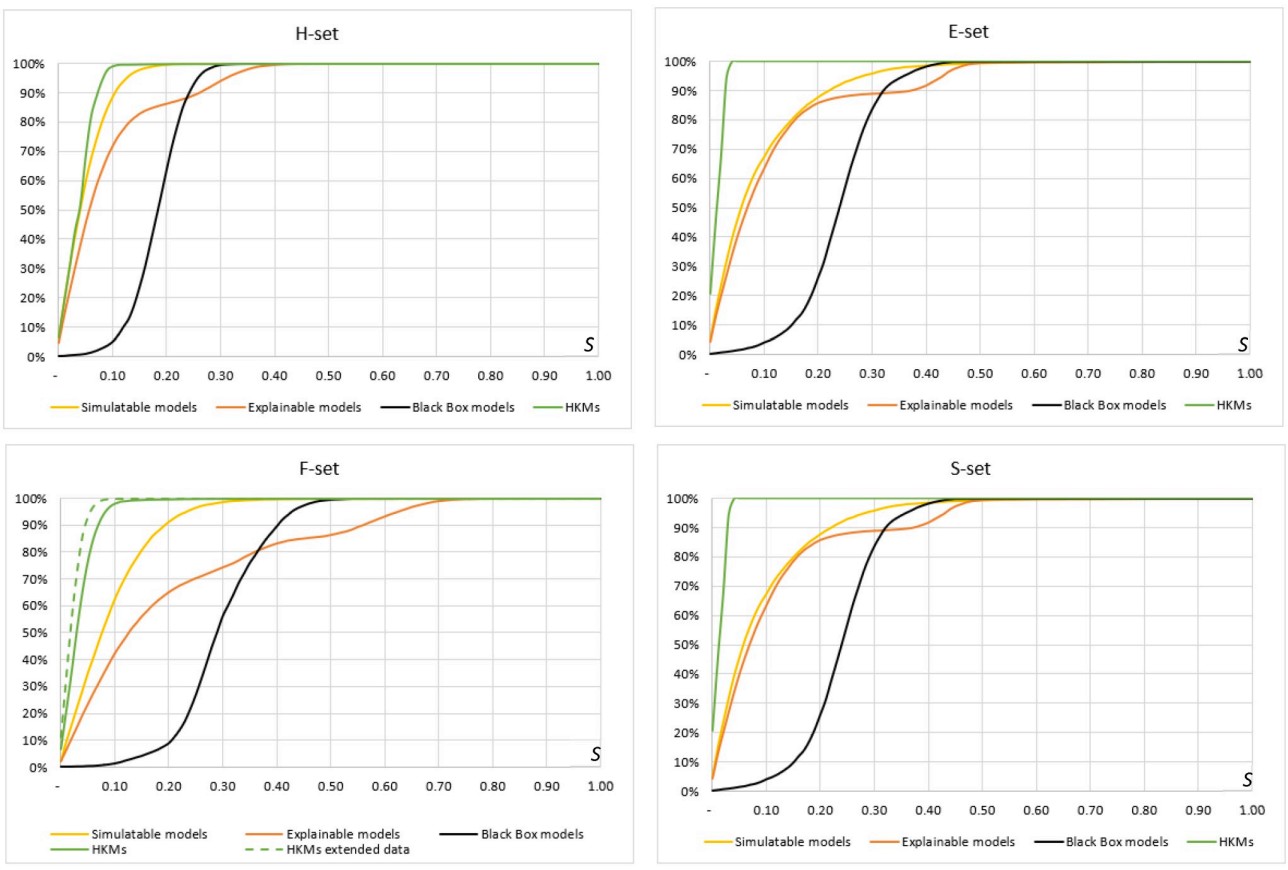

**Fig 4. Model stability P(S) as a function of model quality threshold S (S varying from 0 to 1 in $F_1$ metric).**

Another very important–and less obvious–advantage of the HKM models originates from the exhaustive nature of the HKM learning approach postulated earlier. Learning multiple nearly optimal HKMs helps identify the classes of optimal decisions, which in turn can provide more knowledge about the data (for example, identifying different patient phenotypes for alternative treatments). Additionally, this best model *multiplicity* helps select the models with the lowest application cost: for instance, selecting only the HKMs with the most practically accessible features.

Fig 5 provides a good illustration with a few top COVID-19 HKMs, trained to identify the patients with the highest risk of ICU oxygen support. In this case, multiple models reflect different important classes of high-risk patients (with connective tissue disorder, leukemia, high body mass index, age, etc.), requiring elevated attention. In classical ML, these important rules, even if found, would remain invisible, hidden in aggregated ensemble models or neural net layers. With human-like learning, each of these rules is extremely important on its own, improving the overall process understanding. In addition to being interpretable, these models are very easy to remember, and to apply. In essence, they incorporate invaluable clinical knowledge, which could have taken months to acquire by the usual human trial-and-error but took only a few hours to discover with HKM learning. In many areas, this can save not only time and resources, but lives.

## Limits of human-like learning

The study of HKMs introduces another interesting and new intersection of AI and cognitive sciences: the study of the limits of human-like learning. Despite frequent comparisons between

**SUM**(LowO2ReqFlg=TRUE, AvgReading_Neuts_pct>=83,HasCoronaryArteryDiseaseFLG=0)>=2

(HeartRate>=67 **OR** AvgReading_Neuts_pct>=72) **AND** LowO2ReqFlg=TRUE

(AgeAtArrivalYears>79 **OR** AvgReading_EOS_pct<2) **AND** LowO2ReqFlg=TRUE

**OR**(LowO2ReqFlg=TRUE, AvgReading_Neuts_pct>=83, HasLungCancer=TRUE)

**SUM**(TemperatureFahrenheitNBR<99, LowO2ReqFlg=FALSE,  AvgReading_lymphs_pct>=19,  LastBodyMassIndex<37.47)<3

(AvgReading_Neuts_pct>=76 **OR** HasCoronaryArteryDiseaseFLG=0) **AND** LowO2ReqFlg=TRUE

(AvgReading_PH!=5 **OR** HasConnectiveTissueDisorderFLG=0) **AND** LowO2ReqFlg=TRUE

(HasCOPDFLG=0 **OR** HasChronicLungDiseaseFLG=TRUE) **AND** (LowO2ReqFlg=TRUE **OR** AvgReading_GLOBULIN_gdL=5)

**Fig 5. Examples of COVID-19 HKMs, identifying high risk patients.** The models were learned from a clinical COVID dataset similar to H-set and achieved the same quality as conventional machine learning models.

AI and human intelligence, we know little of how much humans can learn from a certain dataset and how deeply this data can be understood.

HKMs, specifically designed to reproduce human data processing abilities, can advance our understanding of these problems. Moreover, by learning multiple optimal HKMs, we essentially simulate a large number of humans, discovering alternative explanations of a given outcome–when different people can arrive to the same optimal conclusions with entirely different logic. In this pool of human learners, the best HKMs can be viewed as the top "human experts", with the most accurate understanding of the underlying dependencies.

The application of this approach leads to many interesting insights, including a more robust comparison between "black box" AI and human-like learning, as we illustrate in Fig 6. In this case, we used the H-set example to compare the top 2000 "black box" model instances (corresponding to XGBoost, which performed best on this dataset) to a few levels of complexity in human-like learning: the top 2000 HKM models with 2, 3 or 4 features, and all possible one-feature HKMs. For a more objective comparison, the results were visualized using two major quality axes: precision and recall.

Several important observations follow.

First, one can clearly see that at four learning features, suggested by human psychology research, human-like learning modeled by HKMs begins to compete with the quality of "black boxes." Moreover, although HKM quality visibly increases when going from single-feature "HKM size 1" to two or more, it improves only marginally at the larger model sizes ("HM size 4" vs "HKM size 3"). This indicates that within the semantics of proposed features, the intrinsic dataset dimensionality can be limited to 3 or 4, with adding more features producing only a very marginal improvement in interpretation. This manifests one of the of most principal differences between human intelligence and AI: while humans can efficiently learn by finding the best few features, AI takes advantage of greedy "weak learning", aggregating large counts of less influential (and randomly chosen) features to achieve better quality. Weak learning aggregation, however, requires significant volumes of data (to avoid overfitting), and large computing power/memory–none of which human learners can provide.

However, as our examples in Figs 3 and 6 suggest, human-like learning can still be efficient in finding the best few features and short logic rules to explain the outcome. In many practical applications, this human-like learning process alone can warrant a very good result, often approaching that of "black boxes" (Fig 6). This is particularly true for data where the best features have already been discovered with earlier experiments and science, leading to the optimal choice of data-describing language (semantics), and more concise data interpretations. In such

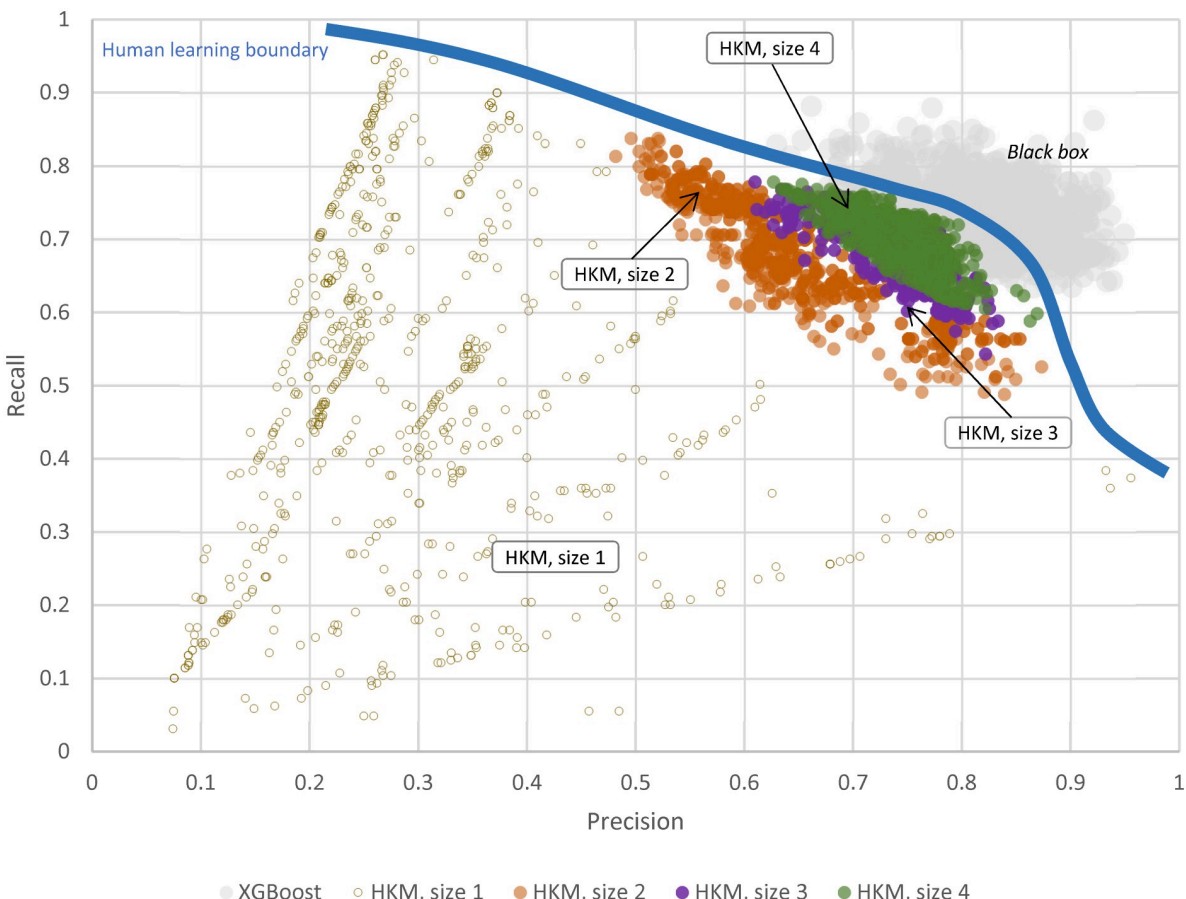

**Fig 6. Using HKM models to visualize the human "comprehensibility" of a specific dataset.** HKM size indicates the number of features used in HKM model learning–from 1 to 4. Size 1 includes all possible one-feature models, sizes 2 and above–only the 2000 best. "Black box" corresponds to the best 2000 instances of the XGBoost model, which outperformed all other models on this dataset.

cases, adding weak learning AI would provide only marginal and practically worthless improvement, aggravated by a significant implementation cost and complete loss of comprehensibility.

When the best features are not yet known, any learning model (HKM or "black box") will be less successful, resulting in relatively low prediction quality. However, even in these cases HKMs can still outperform "black boxes" (such as E-set and S-set examples in Fig 3), meaning that humans, armed with simple optimal decision rules, can outperform AI (likely degraded by mixing semi-informative features with the noise from the rest). Moreover, HKM-learned rules, short and comprehensible, work as engineered features, and can improve our understanding of the hidden data logic. For example, learning that COVID risk increases for patients with pulmonary diseases (Fig 5) provides very practical insights into how COVID is spreading in the human body, instantly leading to better therapy and prevention. Such conceptual breakthroughs cannot be achieved with weak learners, and can originate only from human-like learning, as provided by HKMs.

## HKM computational speed

The time complexity of HKMs with respect to the original variable count $N$, number of thresholds $K$, observation count $L$ and HKM variable count $n$ can be estimated as the product of

several principal components:

$$O(N, K, L, n) = S(N, K, n) \times B(n) \times F(n, L) \tag{1}$$

The first $S(N,K,n)$ term represents the number of n-variable subsets, selected from the $N$ original variables with up to $K$ thresholds per variable. Since $K$ thresholds will produce at most $NK$ Booleans, and HKM constraint $n \leq 4$, this results in

$$S(N, K, n) = \binom{NK}{n} = \frac{(NK)!}{n!(NK - n)!} < \frac{1}{n!}(NK)^n \leq O((NK)^4) \tag{2}$$

The second B(n) term represents the number of Boolean formulas built from n Boolean variables, which is $B(n) = 2^{2^n}$. This makes $B(n)$ the fastest-growing term in Eq 1, but it also becomes the most significantly truncated by the HKM constraint of $n \leq 4$: we found that fewer than 100 Boolean expressions could be made that satisfy the HKM definition; therefore, we can limit B(n) by this constant.

The third F(n, L) term is the time complexity of computing a Boolean expression of n Boolean vectors, L observations in each, which must be done for each (formula, variable subset) pair. Therefore, with the same bound on n, we can compute F(n, L) in O(L), if not faster (using recursive techniques).

Thus, HKM time complexity can be estimated as

$$O(N, K, L, n) = O(L(NK)^n), \quad n \leq 4 \tag{3}$$

Since HKM expressions are small and hard to overfit, we can always reduce the impact of high observation count $L$ by subsampling the data. Therefore, HKM time complexity becomes polynomial with respect to the number of original variables N, and thresholding count K.

This relatively low complexity, combined with very time-efficient Boolean math, and constant memory (as long as it can hold small sets of $n$ Boolean variables) makes HKMs surprisingly attractive for a brute-force algorithm. From our numerical experiments, we were able to process up to $10^8$ subsets on a Xeon multicore workstation in 2–3 days, using suboptimal interpretable programming languages (Matlab, Python) with parallel processing and vectorized Boolean math. For n = 4, processing $10^8$ subsets is equivalent to NK≈200, or N≈20–25 (assuming K<10, and some of N variables being Boolean already); for n = 3, we can increase N to 80–85. This rather high N proves to be sufficient for many real-life datasets, thus making HKM a very practical alternative to classical machine learning and human trial-and-error. When even higher N or K are needed, the computational speed can be increased with more efficient compiled code (such as C++), algorithm optimization (such as branch-and-bound), reducing threshold count $K$, and deterministic greedy searches (such as forward selection, or combining smaller-size HKMs into large-size HKMs).

Finally–and probably most importantly–the most impressive time savings are achieved by the HKM approach itself rather than the Boolean search time. Once HKMs are discovered, they are applied and computed by their human users, with absolutely no need for any computing time or power. Thus, by investing more time into the model discovery, we save colossal resources for the subsequent model use.

## Limitations and applicability

Any approach to human knowledge models, including ours, will be naturally limited by the "human factor", where human ability to reason and to "compute" is significantly affected by numerous aspects such as stress and emotions [42], cognitive bias [43, 44], and even sense of

hunger [45]. However, these limitations are universal, and will impact all areas of human activity, including interaction with other forms of AI. Constraining HKMs to the simplest possible expressions can help keep these problems at a minimum, and we intentionally formulated our HKM constraints based on the most average human performance.

It is also understandable that HKMs will fail in many cases–for instance, in problems with high decision-making branching factors (such as chess or formal proofs)–where small, constrained logic will not capture the essence of the problem, and substantial memory will be required to process all possible scenarios. Nonetheless, in many cases HKMs still can become a good starting point, to eliminate at least the most obvious blinders, which is still extremely important in many applied areas (such as healthcare).

Finally, HKMs will be limited by our human ability to formulate, to observe, and to measure the most important features. A good illustration of this includes many image-recognition problems, where the principal advantage of deep-learning "black boxes" lies in their ability to capture complex features through image pixel convolutions–a task impossible for a human. However, as our human image-recognition experience confirms, at least some of these visual features can be expressed in humanly-computable ways, which can then lead to successful HKM development. We believe that this presents a very interesting direction for future research.

## Conclusions

In this study, we introduce the class of Human Knowledge Models (HKMs), as an attempt to formalize the set of functions that humans can efficiently compute without any additional resources. Based on earlier research in human psychology and cognition, HKM formalization achieves several principal goals:

- It reduces potentially complex Boolean expressions to the shortest possible form, to ensure that they can be efficiently processed by humans

- It encapsulates applied human knowledge in the mathematical form of an HKM machine learning model, which can be trained to discover human decision logic directly from the data.

- It improves the previous approaches to "interpretable" and "explainable" AI by targeting active forms of human decision-making and provides a more formal and accurate definition of this functionality.

- It offers a new and very capable class of models, which can successfully compete with the classical "black box" machine learning, thus significantly broadening the scope of applied AI into the areas where "black boxes" cannot be accepted.

- It helps us understand the limits of human-like learning from data.

Most importantly, we consider HKMs as the first successful approach to closing the "human knowledge" feedback loop. While human knowledge leads to collecting more experimental data, the data processed by contemporary "black box" AI does not feed back into advancing human knowledge. With HKMs, we still rely on excessive computing power, but we use it in a completely different way: to discover the best human reasoning logic (instead of incomprehensible numerical output). Therefore, we believe that further refinements in HKM formalization and learning algorithms will present extremely valuable directions in both scientific and practical AI applications. This can be particularly useful in many applied fields–such as medicine, healthcare, ecology, and software development–where large data volumes need to be understood and transferred into better human-usable decision-making logic.

## Supporting information

**S1 File.**
(DOCX)

**S1 Data.**
(XLSX)

## Acknowledgments

Sergei O. Kuznetsov would like to thank HSE University Basic Research Program for their support.

## Author Contributions

**Conceptualization:** Oleg S. Pianykh.

**Data curation:** Ilia Semenkov.

**Formal analysis:** Egor Dudyrev, Gleb Gusev, Oleg S. Pianykh.

**Investigation:** Egor Dudyrev, Sergei O. Kuznetsov, Andrew Sharp, Oleg S. Pianykh.

**Methodology:** Sergei O. Kuznetsov, Gleb Gusev, Andrew Sharp, Oleg S. Pianykh.

**Project administration:** Sergei O. Kuznetsov, Oleg S. Pianykh.

**Resources:** Ilia Semenkov, Sergei O. Kuznetsov, Gleb Gusev.

**Software:** Ilia Semenkov, Andrew Sharp.

**Supervision:** Sergei O. Kuznetsov, Oleg S. Pianykh.

**Validation:** Egor Dudyrev, Ilia Semenkov, Gleb Gusev, Andrew Sharp.

**Visualization:** Egor Dudyrev, Andrew Sharp.

**Writing – original draft:** Egor Dudyrev, Ilia Semenkov, Oleg S. Pianykh.

**Writing – review & editing:** Ilia Semenkov, Sergei O. Kuznetsov, Gleb Gusev, Andrew Sharp, Oleg S. Pianykh.

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
