## [Decision Letter · Decision Letter 0]

8 Aug 2022

PONE-D-22-19468Human Knowledge Models: Learning Applied Knowledge from the DataPLOS ONE

Dear Dr. Pianykh,

Thank you for submitting your manuscript to PLOS ONE. After careful consideration, we feel that it has merit but does not fully meet PLOS ONE’s publication criteria as it currently stands. Therefore, we invite you to submit a revised version of the manuscript that addresses the points raised during the review process.

We look forward to receiving your revised manuscript.

Kind regards,

Sathishkumar V E

Academic Editor

PLOS ONE

Journal Requirements:

Reviewers' comments:

Reviewer's Responses to Questions

**Comments to the Author**

1. Is the manuscript technically sound, and do the data support the conclusions?

Reviewer #1: Yes

Reviewer #2: Yes

Reviewer #3: Yes

2. Has the statistical analysis been performed appropriately and rigorously? 

Reviewer #1: Yes

Reviewer #2: Yes

Reviewer #3: Yes

3. Have the authors made all data underlying the findings in their manuscript fully available?

Reviewer #1: Yes

Reviewer #2: Yes

Reviewer #3: Yes

4. Is the manuscript presented in an intelligible fashion and written in standard English?

Reviewer #1: Yes

Reviewer #2: Yes

Reviewer #3: Yes

5. Review Comments to the Author

Reviewer #1: In this paper the authors introduce the class of Human Knowledge Models (HKMs), as an attempt to formalize the set of functions that humans can efficiently compute without any additional resources. The contributions are novel and the organization is good. However, the following minor corrections to be done before consider this paper for publications.

1. The Introduction is small, and it do not clearly explain the motivation. It is recommended to provide the challenges of the existing models and motivation clearly.

2. The references considered for this work are older. Consider the state-of-the-art papers for references.

3. Some of the references are available in the existing papers.

4. The results are well presented, but the reasons for achieving the better performance of proposed work over the existing approaches to be discussed.

Reviewer #2: The authors present an interesting manuscript on human knowledge models in AI. Study design and research questions are clearly described. In this sense, it is easy to understand the aim of this study. The bright side of the manuscript is that to provide some useful practical details on related topic. In this context, the study contributes to different fields. The authors explained their methods, and results but some sentences of the paper manuscript (highlighted the pdf file) are too long and there are some grammatical mistakes. Therefore, I would like to make some minor suggestions to improve the quality of the paper as below:

In general, methods and results should be written in past tense but some in some parts of the manuscript some sentences were written in simple present tense.

In methods and results sections, authors used “As a result” in several times (highlighted the pdf file) and this disrupts the flow of the subject and the continuity of the reading. Instead of this using this term, authors should clearly refer the related objective/results or method.

Manuscript section names and reference styles should be corrected by the journal guidelines.

Abstract

Abstract should be one paragraph, please unite two separate paragraphs.

we formalize -> we formalized

We use -> we used

Our results prove -> Our results proved

How the results contribute to further studies? A few words about to whom the results will be important will be valuable. In my opinion, it is always good to finish the abstract with such a sentence.

Introduction

-Authors defined the purpose of the work and its significance in introduction section. However, introduction should briefly place the study in a broad context and highlight why it is important. In this context, in my opinion, introduction should be started with a paragraph that defines the importance of AI/machine learning algorithms/models which benefits the various research fields by giving several examples from different fields (e.g. ecology, healthcare, software development etc.). Then, authors can continue with explaining the limitations of these AI/machine learning algorithms/models. In this way, bridge between the problem and objectives of the study would be stronger.

For example, Akçay et al. (2020), found that computer-aided counting outperformed the manual counting with respect to both accuracy and time (doi: 10.3390/ani10071207). There are many examples available from different fields.

In other words, authors should mention that “yes AI or machine learning models help/contribute researchers to archive their tasks and/or solve problems in less time but the models have limitations and need to be improved”.

-The following sentence should not be in introduction section. Authors may add this sentence to abstract as a last sentence or results section.

“As a result, our study reveals the great potential of HKMs, particularly in the decision-making applications where “black box” models cannot be accepted. Moreover, this improves our understanding of how well human decision-making, modelled by HKMs, can approach the ideal solutions in real-life problems.”

Conclusion

A few words about future work and to whom the results will be important will be valuable. Authors may say with a sentence that their findings contribute the different fields such as medicine, healthcare, ecology, software development etc…

Also, I made some highlights on the manuscript pdf file.

Reviewer #3: The article proposes a new model (HKM). The work seems to be very interesting. The following are the queries are from my side:

1. In what way, does HKM differ from Reinforcement learning?

2. What is the rationale behind choosing 4 features and how are these 4 features chosen?. The number of features varies across the datasets.

3. if possible, provide a quantitative tool for measuring the performance of HKM?

4. even though the authors used the data as such to maintain the originclaity, will not have any impact on the performance?

5. M best HKM models of n variables are produced as output? How do they compare to each other?

6. Whether the authros tested HKM models by random querying?

7. Whether the models generated for one dataset can be used for another? That, is it possible to transfer the knowledge gained? Or Update the knowledge when the dataset is added with new features.

8.

6. PLOS authors have the option to publish the peer review history of their article (what does this mean?). If published, this will include your full peer review and any attached files.

Reviewer #1: No

Reviewer #2: No

Reviewer #3: No

---

## [Author Response · Author response to Decision Letter 0]

17 Sep 2022

Our detailed responses to reviewers are uploaded in a separate file

---

## [Decision Letter · Decision Letter 1]

26 Sep 2022

Human Knowledge Models: Learning applied knowledge from the data

PONE-D-22-19468R1

Dear Dr. Pianykh,

We’re pleased to inform you that your manuscript has been judged scientifically suitable for publication and will be formally accepted for publication once it meets all outstanding technical requirements.

Kind regards,

Sathishkumar V E

Academic Editor

PLOS ONE

Additional Editor Comments (optional):

Reviewers' comments:

Reviewer's Responses to Questions

**Comments to the Author**

1. If the authors have adequately addressed your comments raised in a previous round of review and you feel that this manuscript is now acceptable for publication, you may indicate that here to bypass the “Comments to the Author” section, enter your conflict of interest statement in the “Confidential to Editor” section, and submit your "Accept" recommendation.

Reviewer #1: All comments have been addressed

Reviewer #3: (No Response)

2. Is the manuscript technically sound, and do the data support the conclusions?

Reviewer #1: Partly

Reviewer #3: (No Response)

3. Has the statistical analysis been performed appropriately and rigorously? 

Reviewer #1: Yes

Reviewer #3: (No Response)

4. Have the authors made all data underlying the findings in their manuscript fully available?

Reviewer #1: Yes

Reviewer #3: (No Response)

5. Is the manuscript presented in an intelligible fashion and written in standard English?

Reviewer #1: Yes

Reviewer #3: (No Response)

6. Review Comments to the Author

Reviewer #1: The authors addressed all the recommended comments, and this version is improved well over the previous version. So, I am recommending this paper for publication in this journal.

Reviewer #3: (No Response)

7. PLOS authors have the option to publish the peer review history of their article (what does this mean?). If published, this will include your full peer review and any attached files.

Reviewer #1: No

Reviewer #3: No

---

## [Editor Report · Acceptance letter]

11 Oct 2022

PONE-D-22-19468R1 

Human Knowledge Models: Learning applied knowledge from the data 

Dear Dr. Pianykh:

I'm pleased to inform you that your manuscript has been deemed suitable for publication in PLOS ONE. Congratulations! Your manuscript is now with our production department. 

Kind regards, 

on behalf of

Dr. Sathishkumar V E 

Academic Editor

PLOS ONE